# Unraveling the structure and role of Mn and Ce for NOx reduction in application-relevant catalysts

Lieven E. Gevers[1], Linga R. Enakonda[1], Ameen Shahid [1], Samy Ould-Chikh [1], Cristina I. Q. Silva[1], Pasi P. Paalanen[1], Antonio Aguilar-Tapia [2], Jean-Louis Hazemann [2,3], Mohamed Nejib Hedhili[4], Fei Wen[5] & Javier Ruiz-Martínez [1✉]

Mn-based oxides are promising for the selective catalytic reduction (SCR) of NOx with $NH_3$ at temperatures below 200 °C. There is a general agreement that combining Mn with another metal oxide, such as CeOx improves catalytic activity. However, to date, there is an unsettling debate on the effect of Ce. To solve this, here we have systematically investigated a large number of catalysts. Our results show that, at low-temperature, the intrinsic SCR activity of the Mn active sites is not positively affected by Ce species in intimate contact. To confirm our findings, activities reported in literature were surface-area normalized and the analysis do not support an increase in activity by Ce addition. Therefore, we can unequivocally conclude that the beneficial effect of Ce is textural. Besides, addition of Ce suppresses second-step oxidation reactions and thus $N_2O$ formation by structurally diluting MnOx. Therefore, Ce is still an interesting catalyst additive.

[1] King Abdullah University of Science and Technology, KAUST Catalysis Center, Catalysis Nanomaterials and Spectroscopy (CNS), Thuwal 23955, Saudi Arabia. [2] Institut de Chimie Moléculaire de Grenoble, UAR2607 CNRS Université Grenoble Alpes, F-38000 Grenoble, France. [3] Institut Néel, UPR 2940 CNRS, F-38042Grenoble cedex 9, Grenoble, France. [4] King Abdullah University of Science and Technology, KAUST Core Labs, Thuwal 23955, Saudi Arabia. [5] Umicore AG & Co. KG, Rodenbachen Chaussee 4, 63457 Hanau-Wolfgang, Germany. ✉email: Javier.ruizmartinez@kaust.edu.sa

The selective catalytic reduction (SCR) of environmentally harmful nitric oxide (NO) with ammonia ($NH_3$) is a well-known and established technology for the denitrification of exhaust gases from stationary (power plant) and mobile (e.g., lean-burn engines) sources[1–3]. However, the more stringent global legislations and the relatively low exhaust temperatures of more efficient engines and low-load engine operations require the search for more efficient catalytic systems. For example, In Euro 6 stage, the European Union legislative authorities have tightened the limits of nitrogen oxides being emitted from diesel cars (from 180 mg $NO_x$/km in Euro 5 to 80 mg $NO_x$/km in Euro 6)[4]. A wide variety of catalytic systems based on metal-containing zeolites and mixed metal oxides have been investigated in this reaction. The introduction of Cu-exchanged small pore molecular sieves such as Cu-SSZ-13 and Cu-SAPO-34 have been a revolutionary technology for SCR applications[5] and have an optimal performance between 200–450 °C[6–8]. Among mixed metal oxides, $V_2O_5$-$WO_3$-/$TiO_2$ catalysts give >90% NO conversion at gas hourly space velocities (GHSV) of 60000–90000 h$^{-1}$ between 250–400 °C[9–13]. However, all these systems fall short of providing sufficient performance at temperatures below 200 °C. Catalysts operating at lower temperatures are imperative in mobile applications due to engine cold start[14] and new advances in low-temperature combustion[15]. In this respect, manganese-containing mixed metal oxides exhibit excellent catalytic activity in the $NH_3$-SCR reaction operating at temperatures below 200 °C, and therefore is of particular interest as a potential low-temperature component in $NH_3$-SCR[16–23].

Typically, Mn-based catalysts are prepared by impregnation or homogeneous precipitation methods with other metal oxides, such as Ti and Ce oxides, that act as support, dopants, or promoters. During the last decades, the role of the different components on the catalytic activity and selectivity have been debated extensively[3]. Mn catalytic activity originates from its excellent redox ability at low temperatures. The importance of specific surface area, dispersion, and oxidation state of the different Mn oxides have been highlighted[24–26]. $TiO_2$ is considered a metal oxide support providing optimal dispersion of Mn active species, surface area, thermal stability, and Lewis acid sites to adsorb $NH_3$[27,28]. For Ce and other transition metal, there is no clear consensus on their role on the catalytic reaction. The promotional effect is often explained by an improvement of the catalytic redox cycles by intimate contact of the active Mn oxides and the promotors[29–32]. Among the transition metals, Ce is widely used and probably one of the most promising promotors[3]. In binary MnCe systems, the addition of Ce was reported to improve the conversion levels compared to individual Mn oxides[33,34]. This promotional effect is generally explained by an enhancement of the redox functionality, which is proven by the easier reduction of Ce and/or Mn during temperature-programmed reduction experiments[35]. Baiker et al. also postulated that binary MnCe oxides have a higher adsorption of NO and $NH_3$, which promotes catalytic activity[36]. In ternary MnCeTi oxides, the improvement of activity by Ce is also frequently explained by an increase of the Mn redox properties[35,37–39]. In contrast, other studies suggest that the MnCe electronic interaction decrease the activity of Mn oxide species for NO conversion[40] by a reduction of the $Mn^{4+}$/$Mn^{3+}$ ratio. On the basis of the measured surface areas, binary MnCe[33,34,36] and tertiary MnCeTi[35,37,40–42] systems show better textural properties when Ce is added, but this is rarely discussed as a main promoting effect.

To resolve this unsettled dilemma, we studied the structure and catalytic performance of Mn, Ce, and Ti mixed-oxide catalysts with a wide range of metal-oxide compositions. The catalysts were prepared by a homogeneous precipitation method designed to obtain mixed oxides with amorphous structure and a homogeneous dispersion of the distinct metal oxides. More specifically, 30 catalysts were systematically synthesized with different Mn, Ce, and Ti compositions. To verify the formation of an amorphous and homogeneous mixed oxide, we have chosen a multi-technique approach combining XRD, electron microscopy and high-resolution X-ray photoelectron spectroscopy. The low-temperature $NH_3$-SCR performance of all the catalysts was investigated under relevant conditions encountered in a car exhaust (see experimental details).

## Results

**Catalyst synthesis, structure, and metal-oxide spatial distribution.** After synthesis of the catalysts, we aim to characterize their structure and chemical properties by a multi-pronged approach. The crystalline structure of the catalysts was studied by powder X-ray diffraction, and crystallite sizes were calculated from diffraction peaks integral breadth using the Scherrer equation. The results are summarized in the ternary diagram plotted in Fig. 1 and the detailed diffraction patterns and crystallite size analysis are in the Supplementary information (Supplementary Fig. 2, Supplementary Fig. 3, and Supplementary Table S2).

The binary catalysts have a certain degree of crystallinity and show reflection lines of fluorite $CeO_2$ structure in the CeTi and MnCe oxides and rutile $TiO_2$ in the MnTi oxides. $MnO_x$ crystalline phases, α-$Mn_2O_3$ (Bixbyite), $Mn_5O_8$, $Mn_3O_4$ (Hausmannite), and MnO(OH) (Groutite) are observed on the MnCe oxides, whereas binary MnTi samples display weak and broad reflections from MnOx phases, indicating nano-crystallites below 3–4 nm, most likely on the $TiO_2$ surface. The existence of mono-component metal oxide crystalline phases is an indisputable proof that the preparation method is not successful for the synthesis of well-mixed oxides in binary systems. In sheer contrast to the binary systems, the diffraction patterns of most of the ternary systems show featureless diffraction patterns indicating the amorphous nature of the samples (Supplementary Fig. 3). Only a few samples with high Ti ($Mn_{0.08}Ce_{0.13}Ti_{0.79}$) or Ce ($Mn_{0.07}Ce_{0.55}Ti_{0.37}$ and $Mn_{0.11}Ce_{0.48}Ti_{0.41}$) content display reflections from anatase $TiO_2$ or fluorite $CeO_2$, respectively (see Supplementary Fig. 4). This suggest that the third metal

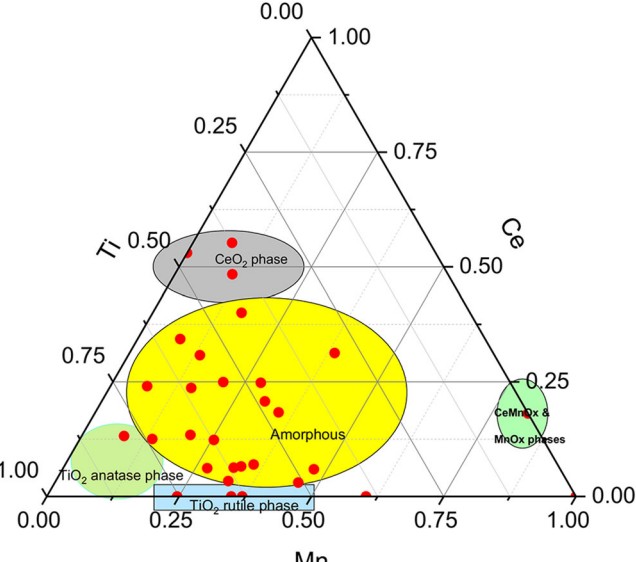

**Fig. 1 Crystallinity of mixed oxides.** Ternary phase diagram of the synthesized MnCeTiOx catalyst samples. The colored areas highlight groups of samples with similar crystal phases, measured by XRD.

component facilitates the formation of a ternary amorphous phase, which is the first step on obtaining a homogeneous mixed oxide. Very importantly, the effect of Ce is crucial in the formation of an amorphous phase, as small amounts of such a component inhibits the formation of crystalline phases, as can be observed in the low edge of the ternary diagram in Fig. 1.

The bulk and surface composition of the catalysts were compared by using Inductive Coupled Plasma (ICP) and high-resolution X-ray Photoelectron Spectroscopy (XPS), respectively (Supplementary Table S1). The bulk chemical composition of the catalysts is comparable with the theoretical composition, with a slight deviation to lower Mn concentrations, which suggests that not all the Mn is precipitated during synthesis. One plausible explanation is the formation of soluble Mn complexes with ammonium ligands, which are less prone to hydrolysis-condensation reactions[43]. Those are expected to remain in the supernatant during the washing by centrifugation, thus decreasing the final Mn content on the catalyst. The bulk composition was compared to the surface composition of selected samples by XPS. For the MnTi binary, there is an enrichment of Mn on the surface, in line with the hypothesis that Mn is supported on $TiO_2$. For the ternary systems, the results show that surface compositions are similar to bulk compositions, with some of the samples displaying a modest enrichment of Mn and Ce on the surface.

To further investigate the structure, metal oxide distribution, and local composition of the catalysts, we performed a high-resolution electron microscopy study. A representative transmission electron microscopy (TEM) image of a binary MnTi Sample ($Mn_{0.37}Ce_{0.00}Ti_{0.63}$) is shown in Supplementary Fig. 5. We observed nanosized and crystalline $TiO_2$ particles decorated with layers of amorphous material. High-angle annular dark-field (HAADF) STEM imaging and elemental mappings computed from energy-dispersive X-ray spectroscopy (EDX) data, presented in Supplementary Fig. 6, reveal that Mn is well dispersed on the $TiO_2$ particles. In contrast, TEM images of a representative MnCeTi ternary system, in Fig. 2a, b, corroborate the existence of a purely amorphous system. The structural features resemble an agglomeration of shapeless nanoparticles forming a random porous structure. Annular dark-field (ADF) STEM imaging and elemental mapping computed from EELS data confirm the homogeneous distribution of Mn, Ce, and Ti metals over individual catalyst particles.

All the results strongly confirm that our synthesis route renders a homogeneous distribution of all the metal atoms in the ternary system and indicate that our synthesis method is effective for the preparation of ideal perfectly-mixed ternary metal oxides. From the characterization measurements conducted, we illustrate the structure of the MnTi binary and the MnCeTi ternary catalysts in Fig. 3.

**Impact of metal oxide composition on catalyst textural properties, oxide reducibility, and speciation**. The specific surface areas of the discrete catalysts are presented in the ternary diagram in Fig. 4 and in Supplementary Fig. 7 and Supplementary Table S3.

The catalyst materials have a clear type IV isotherm, with an H2 or H3 loop characteristic of mesoporous coming from the agglomeration of the metal-oxide nanoparticles[44]. Clearly, Ce content has a significant impact on specific surface areas of the ternary catalysts. The addition of Ce plays a key role as structural promotor increasing the surface area from 108 to 245 $m^2/g$ when Ce content increases from 0 to 20 mol%. A further increase in Ce has a negative impact and surface areas drop down to 62 $m^2/g$ when Ce concentration reaches 55 mol%, similar to previous studies on CeTi[45], MnCe[46] binary systems and MnCeTi ternary

systems[47]. Comparing the XRD and BET data, the increase of surface area is strongly correlated with the formation of amorphous structures.

To investigate the redox properties of the materials, temperature-programmed reduction (TPR) experiments with $H_2$ were performed. Overall, the reduction peak at 200–450 °C is attributed to the reduction of $Mn^{4+}$ and $Mn^{3+}$ species to $Mn^{2+}$ [48,49], whereas the main reduction peak at 550–650 °C is ascribed to the reduction of $Ce^{4+}$ to $Ce^{3+}$ in a mixed oxide phase[50–52]. The difference in the TPR of pure $CeO_2$, plotted on supplementary Fig. 8, confirms that Ce is well-mixed and strongly interacting with Mn and Ti. The samples containing a low Mn/Ce molar ratio, plotted in Fig. 5a, show a monotonic decrease in the reduction temperature of Ce when Mn content increases. According to literature reports, the main parameters influencing the reduction temperature are specific surface area and the presence of other metals in intimate contact[53]. In this case, the surface area of the binary Ce-Ti system is higher than the Mn-contained samples shown in Fig. 5a. Therefore, changes in the reduction temperature are unrelated to specific surface area and can be rationalized as a consequence of the close proximity of Mn and Ce species, which improves the reducibility and thus the redox properties of Ce. The TPR of samples containing high amount of Mn are shown in Fig. 5b. For the binary MnTi samples, two main peaks from Mn reduction (289–306 °C and 341–408 °C) are observed. Addition of Ce in the catalyst formulation has a strong impact on the TPR profiles: the low-temperature peak strongly decreases with small amounts of Ce and the high-temperature contribution shifts to higher temperatures with increasing Ce content (see also Supplementary Fig. 9). We surmise that the changes in the TPR profiles probably result from the lower reducibility of the Mn species in close contact with Ce.

To gain more nuanced insight into the effect of Ce to the catalyst properties, a detailed high-resolution XPS study was performed. Oxidation states of Mn were rigorously fitted from a set of Gaussian-Lorentzian components per oxidation state, due to the multiplet splitting between the unpaired electrons in Mn 2+, 3+, and 4+. The set of components of the discrete oxidation states were obtained from measurements of pure MnO, $Mn_2O_3$, and $MnO_2$ oxides and the results were compared with previously reported measurements[54]. More experimental details can be found in the supplementary information and Supplementary Fig. 10. The Mn ($2p_{3/2}$) spectra from selected samples show a high amount of $Mn^{3+}$ species in the binary MnTi catalysts, whereas a combination of $Mn^{4+}$ and $Mn^{2+}$ dominates the spectra of ternary samples (see Supplementary Table S5). There is no clear consensus in the literature on the effect of Ce on the Mn oxidation state. Whilst several authors found an increase in $Mn^{3+}$ species with the addition of Ce[35,39,40], Feng et al. observed a slight reduction in $Mn^{3+}$ [42], and others found no clear correlation[55,56]. The origin of the discrepancy could be related to the complexity on the analysis of the Mn ($2p_{3/2}$) spectra and the different structure of the prepared catalysts. In order to improve the confidence in our results, average Mn oxidation states were calculated from the XPS and TPR and gave comparable results (see Supplementary Table 7), which validates our XPS deconvolution method and confirms that the addition of Ce decreases the average Mn oxidation state. Raman spectra of selected binary MnTi and ternary MnCeTi were also collected and qualitative analysis is in line with the XPS and $H_2$-TPR results (see supplementary information, Supplementary Fig. 11 and Supplementary Table S6). Based on our structural characterization results, we speculate that the reduction in the average Mn oxidation state is due to the existence of Ce species in close proximity to Mn. This might explain the shifting of TPR profile to high temperature after introducing Ce into the MnTi system.

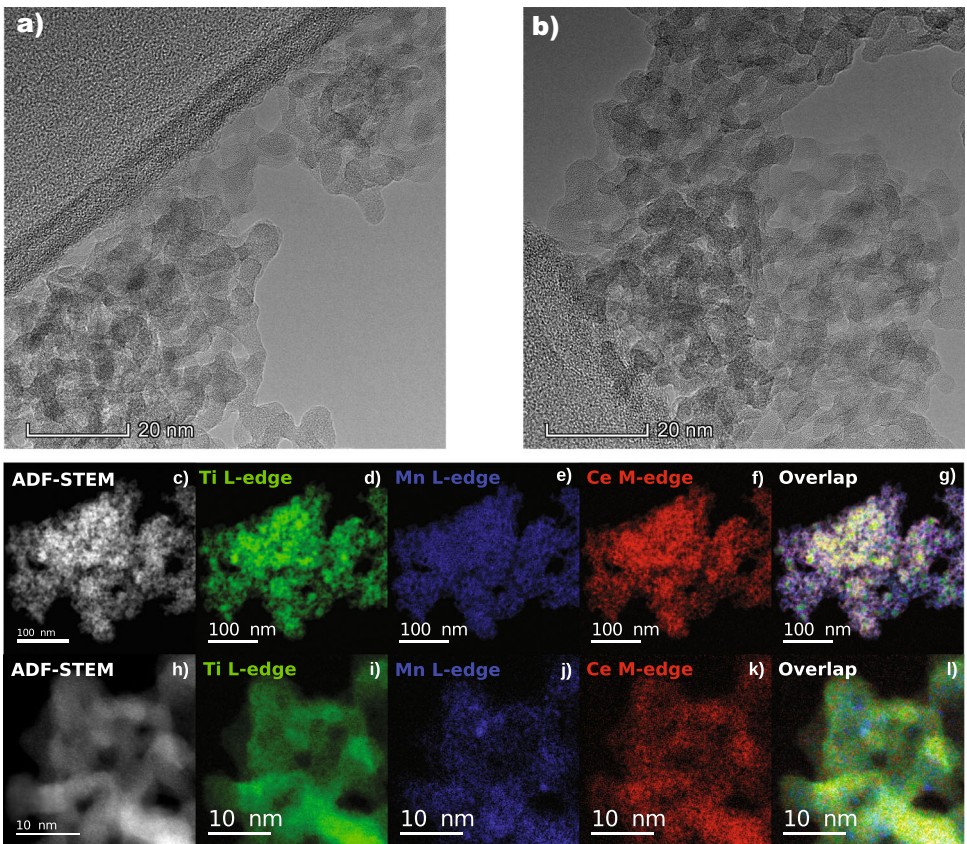

**Fig. 2 Morphological and elemental structure of MnCeTi ternary catalysts. a, b** Representative HRTEM images of the $Mn_{0.14}Ce_{0.13}Ti_{0.74}$ oxide catalyst. ADF-STEM images and elemental mappings of the same catalyst at (**c–g**) low and (**h–l**) high magnification.

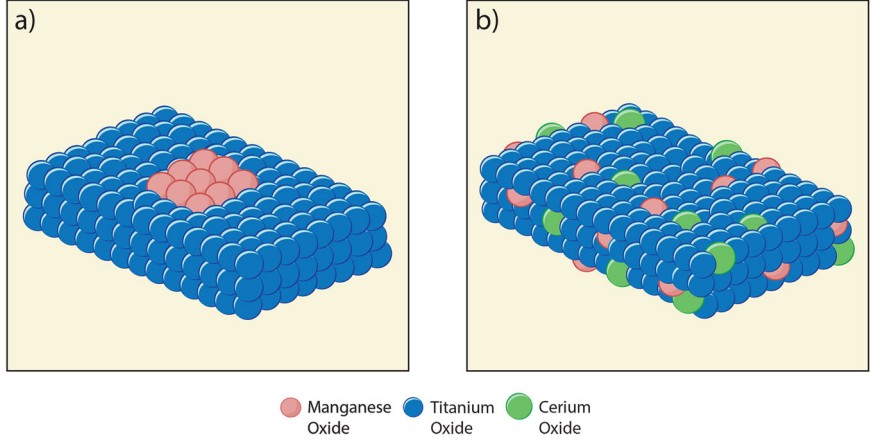

**Fig. 3 Schematic representation of catalyst structures. a** MnTi binary catalysts where amorphous layers of Mn oxide active are on the surface of crystalline $TiO_2$ and **b** MnCeTi ternary catalysts where the metal oxides are amorphous and well mixed.

Additional information about the chemistry and structure of Mn on selected samples was obtained by X-ray absorption spectroscopy (XAS). Mn K-edge spectra were recorded ex situ for selected binary ($Mn_{0.35}Ce_{0.00}Ti_{0.65}$) and ternary ($Mn_{0.37}Ce_{0.04}Ti_{0.60}$ and $Mn_{0.30}Ce_{0.19}Ti_{0.51}$) samples to investigate the effect of Ce in the Mn oxidation state and local structure. All XANES spectra (Fig. 5c) display a weak peak in the pre-edge region (inset) due to the $1\,s \rightarrow 3d$ quadrupolar transitions and a white line characteristic of an oxidized state of Mn sitting mostly in octahedral environments[57,58]. Depending of the Ce concentration, the position of the $1\,s \rightarrow 3d$ transitions are located at different energies indicating different manganese oxidation states. Without Ce, Mn is found in its most

oxidized state ($1\,s \rightarrow 3d$: 6542.0 eV), while adding Ce reduces the average Mn oxidation state. The same findings are observed regarding the absorption edge energies at 6551.6, 6549.2, 6550.0 eV respectively for $Mn_{0.35}Ce_{0.00}Ti_{0.65}$, $Mn_{0.37}Ce_{0.04}Ti_{0.60}$ and $Mn_{0.30}Ce_{0.19}Ti_{0.51}$ compositions. An estimate of the average Mn oxidation state was performed by comparison with manganese oxide standards. More details about the analysis can be found in the supplementary information. Summarizing, the average oxidation states (in Supplementary Table S7) are in agreement with the XPS and $H_2$-TPR data.

Subsequent qualitative assessment of the EXAFS spectra in Fig. 5d, e shows a main peak at about 1.42 Å attributed to Mn-O

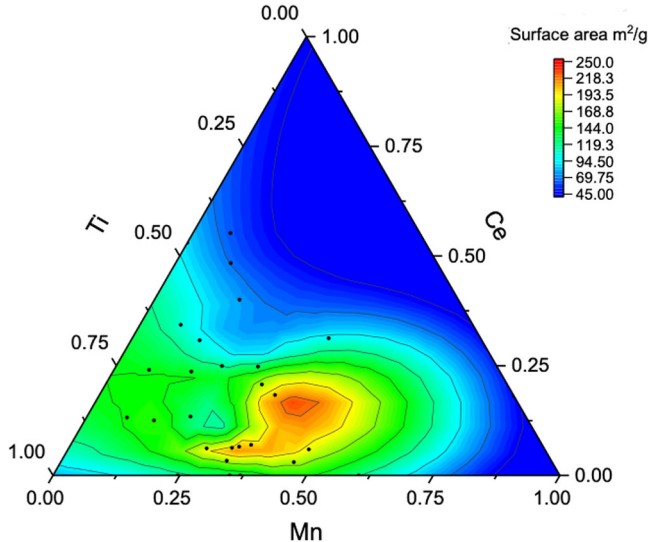

**Fig. 4 Surface area of mixed oxides.** Ternary diagrams showing the Influence of catalyst composition on BET surface area.

scattering paths and several peaks between 2.0 and 5 Å assigned mainly to Mn-Mn paths with some minor contributions from Mn-O scattering and some multiple scattering processes. The peak area corresponding to the Mn-Mn scattering paths are larger for the binary $Mn_{0.35}Ce_{0.00}Ti_{0.65}$ indicating that the later materials has the best long-range order among the three composition (while still appearing amorphous by XRD). The ternary samples display the strongest disorder, supporting than such a system is a homogeneous mixture of mixed oxides.

**Role of Mn and Ce on NO reduction at low temperature.** Next, we inspected the activity of the different samples in the selective catalytic reduction of NO with $NH_3$ at 150 °C. During the activity measurements, there was no $NO_2$ formation, and the only products were $N_2$ and $N_2O$. Activity plot with the amount of Mn shows a close-to-zero intercept of the ordinate, indicating that Mn oxides are the most important species in catalytic performance (Supplementary Fig. 13). Up to 60% of Mn, there is a modest correlation of the increase of activity with the increase in Mn content. However, the results revealed that no evident trend is observed when Mn is above 60% and strongly suggest that Mn content is not the only factor determining catalytic activity.

**Fig. 5 Chemical structure analysis and redox properties of the samples.** Temperature-programmed reductions with $H_2$ of selected samples with (**a**) a low Mn/Ce molar ratio and (**b**) a high Mn/Ce molar ratio. The low-temperature evolution (in purple) is related to the reduction of MnOx phases whereas the high-temperature (in cream) is related to $CeO_2$ reduction. Mn K-edge (**c**) XANES, (**d**) EXAFS, and (**e**) FT-EXAFS spectra of three selected catalyst compositions, $Mn_{0.35}Ce_{0.00}Ti_{0.65}$, $Mn_{0.37}Ce_{0.04}Ti_{0.60}$, $Mn_{0.30}Ce_{0.19}Ti_{0.51}$.

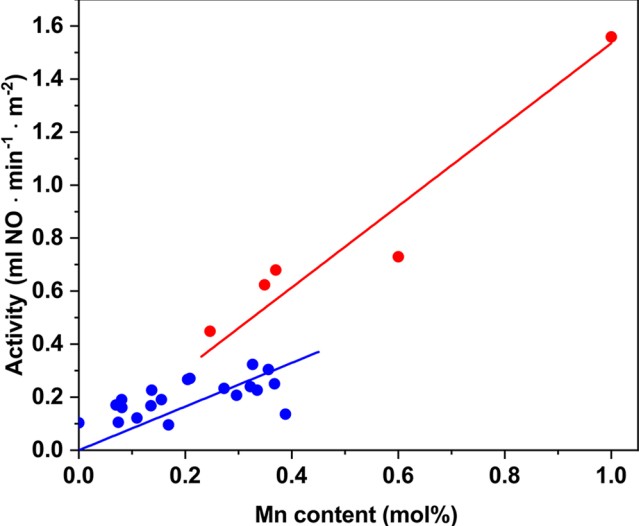

**Fig. 6 Effect of Mn content on catalytic performance.** Surface-specific activities for the NO reduction at 150 °C plotted as a function of the Mn content on the catalysts. The blue dots correspond to the ternary MnCeTi catalysts. The red dots are from the binary MnTi and the individual Mn catalysts. Red and blue trendlines were added to guide the eyes.

To unravel the effect of the distinct metal oxides, we re-examined the catalytic performance by surface area normalization of the activity. Specific activities (ml of NO converted per $m^2$ and per min) were plotted as a function of Mn content in Fig. 6. For the individual Mn and the binary MnTi samples, a clear linear dependency of the measured specific activities with the Mn content is observed. The MnCeTi ternary systems has also a pseudolinear correlation with the Mn content but with a lower slope. The Mn surface enrichment for the MnTi catalysts observed by XPS (Supplementary Table S1) can insufficiently explain the almost twofold increase in specific activity. Therefore, we infer that $MnO_x$ species in the individual Mn and binary MnTi catalysts have similar activity, and those are more active than in the MnCeTi ternary system. The results are also in line with the higher reduction temperature of manganese species observed in the TPR data of MnCeTi ternary systems.

To assess the implication of the different MnOx species on catalytic performance, the normalized activities with the distinct Mn surface species ($Mn^{4+}$, $Mn^{3+}$, $Mn^{2+}$, and total Mn) were constructed in Supplementary Fig. 14. The plots show an increase in activity with the amount of $Mn^{4+}$ and $Mn^{3+}$ species, however there are no direct evidences that the $Mn^{2+}$ species are promoting catalyst activity, in line with previous reports[59]. Hence, the increase in the $Mn^{2+}$ content in the samples containing Ce could, to a certain extent, explain the lower specific activity of those samples. In addition, the substantial effect of Ce on the TPR profiles, shifting reduction peaks of $Mn^{4+}$ and $Mn^{3+}$ to higher temperatures, indicates that other interactions of $Mn^{4+}$ and $Mn^{3+}$ with Ce species can be responsible for the lower specific activity.

We further investigated the nature of the oxygen species in the solid catalyst, which has been suggested to have a significant impact on SCR activity, more specifically on the oxidation reactions[60,61]. The reactivity of oxygen species on the catalyst was investigated by monitoring the formation of $N_2O$ during $NH_3$-TPD experiments. In these experiments, $N_2O$ is formed in the range of 140–40 °C via the oxidation of ammonia with active oxygen species on the catalyst surface. The plots in supplementary Fig. 15. shows that NO activities at 150 °C are correlated to the amount of $N_2O$ evolved during the $NH_3$-TPD experiments,

which points to the direct role of the active oxygen in the overall reaction mechanism at low temperature. Since the NO activity is related to Mn content, we can infer that those active oxygen species are related to Mn species.

The effect of the adsorption of NO and $NO + O_2$ on catalytic performance was also investigated by temperature-programmed techniques in selected samples with similar Mn content (Supplementary Fig. 16). Quantification during NO experiments shows a higher adsorption of NO for the binary MnTi sample. In the case of the $NO + O_2$ adsorption measurements, the sample with a small amount of Ce has a larger amount of NO adsorbed. The lack of a clear trend evidences that the capacity of NO and $NO + O_2$ adsorption is unable to fully describe the catalytic performance.

The effect of Ce on the Mn activity has been previously reported in the literature with inconsistent findings. For example, Liu et al. proposed a positive effect of Ce and Ti in activity by a dual redox cycle consisting of $Mn^{4+} + Ce^{3+} \leftrightarrow Mn^{3+} + Ce^{4+}$ and $Mn^{4+} + Ti^{3+} \leftrightarrow Mn^{3+} + Ti^{4+}$ cycles[35]. A different explanation of the beneficial effect of Ce was proposed by Yang et al., based on a mechanistic study using in-situ FTIR spectroscopy. The authors suggested that MnOx species show a faster rate for the conversion of NO to nitrate or nitrite, whereas $CeO_2$ mainly provides adsorption sites resulting in nitrites species[33]. In line with our results, Wu et al. also observed a negative effect of Ce in the activity in ternary MnCeTi compared to binary MnTi. However, these authors explained the negative effect of Ce by a reduction in the $Mn^{4+}/Mn^{3+}$ ratio[40]. In order to resolve the origin of these contradicting findings, our results were contrasted with previously reported ones after surface-area normalization of the activity data. These data are shown in Supplementary Table S11. The analysis shows that when the activities are surface-area normalized, there is no positive effect of the addition of Ce on the specific Mn activity, which validates our experimental results. Although this analysis is incomplete due to our lack of knowledge on several parameters, such as Mn surface composition, oxidation states, degree of interaction between the different oxide species, etc., there is no apparent trend pointing to an increase in activity by Ce addition. Therefore, we propose that Ce is not improving the intrinsic catalytic properties of Mn species at low reaction temperatures and that the only promotional effect of Ce is purely structural due to an increase in catalyst surface area.

When looking at the catalytic performance at higher temperatures, the role of Ce is clear. Supplementary Fig. 17 shows NOx conversion of selected samples with increasing the amount of Ce. The activity data of all catalyst samples can be found in Supplementary Table S8. The binary MnTi catalysts has the highest activity at low temperatures, but the catalytic performance drastically drops at temperatures >250 °C due to the unselective oxidation of $NH_3$ to NOx. The addition of Ce drops the conversion at low temperatures, but promotes $NO_x$ conversion at temperatures >250 °C, widening the operational temperature window of the catalyst materials. Understanding this effect lies beyond the scope of our investigations as other parameters, such as close proximity of the redox and acidic functions, may govern the reaction at high temperatures[62].

**Role of Mn and Ce on $N_2O$ selectivity at low temperature.** Finally, the $N_2O$ formation at low temperature was investigated on all the samples. Possible effects of conversion on $N_2O$ selectivity were ruled out by inspecting the $N_2O$ selectivity plot as function of conversion in Supplementary Fig. 18. In general, the $N_2O$ formed over $MnO_x$ and binary MnTi catalysts is significantly higher than in the MnCeTi ternary systems. For a

deeper analysis of the selectivity results, the surface-specific formation of $N_2O$ of selected samples was plotted in Fig. 7 as function of the Mn surface density calculated from the XPS measurements. The results clearly show an exponential increase of the $N_2O$ formation at 150 °C with the Mn surface density, indicating that $N_2O$ formation obeys a higher-than-one-order dependence in Mn. Our characterization and $NO_x$ activity measurements pointed to an interaction of Ce with Mn, leading to a decrease in the Mn activity. This can definitely play a role in the lower $N_2O$ activity but does not fully describe the Mn order dependence. A plausible explanation of this behavior is that more than one Mn active species in close proximity must participate in the kinetic formation of $N_2O$. According to literature, two main reaction mechanisms explain the formation of $N_2O$ at low temperature[63]: one base on a Langmuir-Hinshelwood mechanism where NO is oxidized to $NO_3^-$ species that react with $NH_4^+$ to give $N_2O$, and the other is an Eley-Rideal mechanism resulting from the oxidative dehydrogenation of $NH_3$ to NH species that react with NO to yield $N_2O$. A mechanistic study on the

formation of $N_2O$ is beyond the goal of this work, but both proposed mechanisms are based on multistep oxidation of NO and/or $NH_3$. Therefore, we postulate that the formation of $N_2O$ requires the participation of at least two Mn active sites and that is promoted by neighboring Mn active sites. The homogeneous composition of the ternary MnCeTi dilutes the Mn species on the catalyst surface and inhibits subsequent oxidation steps by breaking up the MnOx ensembles. This mechanism is schematically depicted in Fig. 8. Besides the dilution effect, the addition of Ce could also decrease the activity of surface oxygen observed by a suppressed $N_2O$ formation in the $NH_3$-TPD.

To rule out the effect of the amount of ammonia adsorbed on the catalysts on $N_2O$ selectivity, $NH_3$-TPD measurements were performed and the total number of acid sites, obtained from the total amount of ammonia adsorbed, was plotted as a function of Ce content (Supplementary Fig. 19). The total acidity of all the samples was also plotted in the ternary diagram in Supplementary Fig. 20. The total acidity is around 1.1 µmol/$m^2$ up to 20% Ce and then monotonically increases up to around 2.5 µmol/$m^2$. Owing to the fact that the inhibiting of $N_2O$ formation is observed in catalysts with Ce content below 20%, we can exclude the role of the number of acid sites on the $N_2O$ formation.

Concluding, our research results add new insight into our understanding of Mn catalysts for low-temperature $NH_3$-SCR applications and provides a direction toward settling the ongoing debate over the effect of Ce on the Mn active species. We postulate that the activity of the Mn active sites is not positively affected by Ce species in intimate contact. In fact, our results suggest that Ce is decreasing the average oxidation state and activity of Mn active species and is just a structural promotor, increasing catalyst surface area. On the other hand, addition of Ce is increasing $N_2$ selectivity as it suppresses second step oxidation and thus $N_2O$ formation by a dilution effect on the MnOx active sites. The latter still makes Ce an attractive additive for MnTi systems.

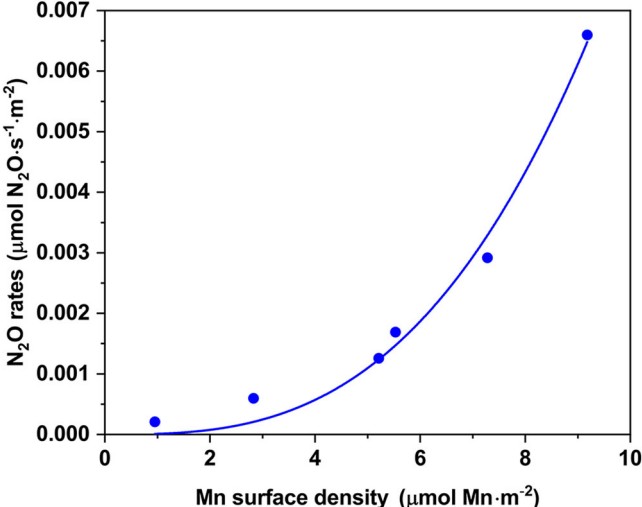

**Fig. 7 Effect of Mn surface density on catalytic selectivity.** Surface-specific $N_2O$ rates at 150 °C as a function of the Mn surface density. Samples from lower to higher Mn surface density: $Mn_{0.08}Ce_{0.55}Ti_{0.37}$, $Mn_{0.14}Ce_{0.13}Ti_{0.74}$, $Mn_{0.21}Ce_{0.25}Ti_{0.54}$, $Mn_{0.30}Ce_{0.19}Ti_{0.51}$, $Mn_{0.36}Ce_{0.04}Ti_{0.60}$, $Mn_{0.35}Ce_{0.00}Ti_{0.65}$. The blue line is a guide to the eye suggesting an exponential-type trend.

## Methods

**Catalyst preparation.** Titanium(IV) sulfate solution ($Ti(SO_4)_2$, Pfaltz & Bauer., 30% in $H_2SO_4$), cerium(III) nitrate hexahydrate ($Ce(NO_3)_3$·6$H_2O$, Sigma-Aldrich, 99.999% trace metals basis), manganese(II) nitrate hydrate ($Mn(NO_3)_2$·x$H_2O$, Sigma-Aldrich, 99.999% trace metals basis), ammonium hydroxide ($NH_4OH$, Alfa Aesar, ACS grade, 28.0–30.0%) were used as received, without further purification.

A series of individual, binary and ternary materials with different molar concentrations were prepared by a controlled co-precipitation method as described in the Supplementary information and Supplementary Fig. 1. Our method is a novel and highly efficient approach, where the aim is to precipitate all metals at the same pH level to obtain a homogeneously well-mixed metal oxide system. This is

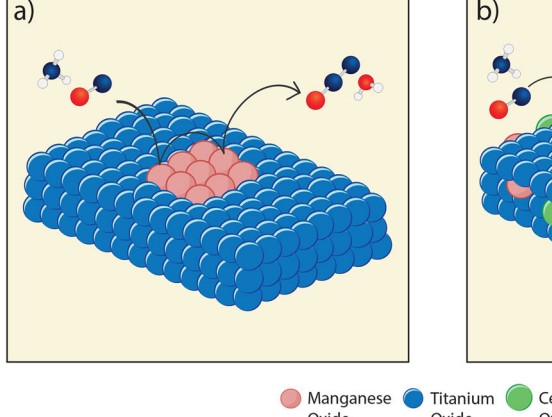
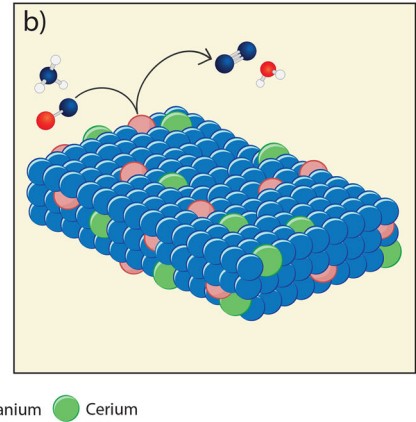

Manganese Oxide — Titanium Oxide — Cerium Oxide

**Fig. 8 Schematic representation of the reaction mechanism. a** MnTi binary catalyst where the close proximity of MnOx species promotes the formation of $N_2O$ and **b** MnCeTi ternary catalyst in which the well-mixed amorphous structure promotes the spacing of MnOx species and therefore reduces the formation of $N_2O$. For the gas molecules: red is oxygen, blue is nitrogen, and white is hydrogen.

done by dual dosing of NH₃ and salt solution at a constant predetermined volumetric ratio. In most literature, the salt solution is added dropwise to an NH₃ solution, but this gives a pH change over time (from high to final low pH) and could lead to a suboptimal co-precipitation of the elements. First manganese nitrate hydrate (Mn(NO₃)₂·xH₂O), and cerium nitrate (Ce(NO₃)₃.6H₂O) were dissolved in deionized water and stirred for 10 min. Then, a 30% titanium sulfate solution [Ti(SO₄)₂ in H₂SO₄] was added to the salt solution. These solutions were mixed under magnetic stirring at a constant speed (400 rpm) for 30 min, leading to a perfectly mixed metal salt solution. The mixed metal salt solution (loaded in a syringe pump) was injected simultaneously along with 14.7 M solution of ammonium hydroxide with a Gilson pump (NH₄OH, Sigma-Aldrich, 97%) to a recipient containing 20 ml of mother solution that is already at the target pH of 10.5. During this simultaneous injection of the metal precursor and base, the resulting suspension was continuously stirred. This procedure allows to operate at a constant pH of around 10.5 by adding the same amount of hydroxide consumed during the catalyst precipitation reaction. The metal oxides will precipitate at the same time, rendering a very high level of homogeneity. Then, the precipitated solution was stirred for 30 min at 400 rpm. The sample was centrifuged at 7500 x g and washed several times with milli-Q water until the conductivity of the supernatant reached to 50 μs.cm⁻¹. Then the samples were dried overnight at 100 °C, followed by calcination at 500 °C for 6 h. The list of samples and compositions is shown in Supplementary Table S1.

**Catalyst characterization.** X-ray diffraction patterns were obtained using a Bruker D8 Advanced A25 diffractometer in Bragg-Brentano geometry with Cu K$_{α, β}$ radiation source operated at 40 kV and 40 mA. β radiation is filtered out with a Ni plate. The diffractograms were measured with step size of 0.05° in the 2θ range of 10–80°. Nitrogen adsorption and desorption isotherms of the samples were measured at 77 K using Micromeritics ASAP-2420 surface area and porosity analyzer instrument. Samples were previously evacuated at 300 °C for 3 h. Specific surface areas and pore size distribution were calculated according to multi-point Brunauer–Emmett–Teller (BET) and Barret–Joyner–Halenda (BJH) method, respectively. From the adsorption data, total pore volumes were estimated at P/P₀ = 0.99. The elemental compositions (Mn, Ce, Ti) of the samples were determined by an inductively coupled plasma spectrometer (Model 8900, Agilent Technologies). The samples were dissolved in HF and HCl. High-resolution Kratos Axis Ultra X-ray photoelectron spectroscopy equipped with a monochromatic Al Kα source was used to determine the surface composition and chemical states of the samples. All analyses were monitored using the C 1 s signal for adventitious carbon (284.8 eV). The chemical states of manganese in the catalysts were determined by peak modeling in CasaXPS software. To model the Mn 2p$_{3/2}$ peaks of the catalysts, pure MnO, Mn₂O₃, and MnO₂ samples were used as reference. Manganese(IV) oxide (99.997% - metals basis) was acquired from Alfa Aesar (Fisher US), manganese(III) oxide (99.9% - trace metals basis) was acquired from Sigma Aldrich, and manganese(II) oxide (99.99% - trace metal basis) was acquired from Acros Organics (VWR). The fitting parameters data (FWHM and Peak positions) obtained from the peak modeling of the standard samples were used for the calculation of the chemical state of manganese in our catalysts. H₂-TPR experiments were performed in Autochem 2950 instrument equipped with a thermal conductivity detector. All catalysts (100 mg) were pretreated in a U-shaped quartz tubular micro-reactor in a flow of Ar at 250 °C for 2 h to yield a clean surface and then cooled down to 40 °C. Then, the temperature was raised from 40 to 1000 °C at a rate of 10 K/min under a flow of 10 vol.% H₂ (90 vol.% Ar). The acidity of samples was determined by temperature-programmed desorption of ammonia (NH₃-TPD). NH₃-TPD of samples was performed in a fixed bed quartz tube reactor. Prior to the measurement, the samples were first pretreated at 500 °C under N₂ flow. The reactor was cool down at 100 °C and samples were saturated with 1050 ppm NH₃ for 30 min. The samples were flush with N₂ for 30 min at room temperature, and then the temperature was increased to 500 °C at a rate of 10 K/min. The outlet gas composition (NH₃, NO, NO₂, N₂O) was monitored by using a MultiGas™ 2030 FTIR Continuous Gas Analyzer. High-resolution transmission electron microscopy (HRTEM) micrographs obtained from a Titan 60–300 TEM (FEI Co, Netherlands) equipped with an electron emission gun operating at 200 kV. The annular Dark-Field scanning transmission electron microscopy (ADF-STEM) in conjunction with electron energy loss spectroscopy (EELS) study was carried out with a Cs-Probe Corrected Titan microscope (Thermo-Fisher Scientific) which was also equipped with a GIF Quantum of model 966 from Gatan Inc. (Pleasanton, CA). STEM-EELS analysis was performed by operating the microscope at the accelerating voltage of 300 kV, using a convergence angle α of 17 mrad and a collection angle β of 38 mrad. Spectrum-imaging dataset includes the simultaneous acquisition of zero-loss and core-loss spectra (DualEELS) using a dispersion of 0.5 eV/channel and were recorded using a beam current of 0.2 nA and a dwell time of 50 ms/pixel. The Ti L$_{2,3}$-edge, Mn L$_{2,3}$-edge, and Ce M$_{4,5}$-edge were selected to build the chemical maps. X-ray absorption spectroscopy (XAS) was performed on the CRG-FAME beamline (BM30B), at the European Synchrotron Radiation Facility in Grenoble. Samples were all diluted with boron nitride (BN) and compressed into a 5 mm diameter pellet to allow the measurement in transmission mode. For all compounds the dilution level corresponded to the optimal sample thickness for transmission experiments (edge jump close to 1). The spectrum of metallic iron was measured with a metallic foil and was also used to

perform the energy calibration of the monochromator (pseudochannel-cut/Si (220), energy resolution 0.335 eV). All XAS data were processed using the FASTOCH package. The XANES and EXAFS spectra were obtained after performing standard procedures for pre-edge subtraction, normalization, polynomial removal, and wave-vector conversion. The NO and NO + O₂ adsorption experiments were conducted in a fixed bed quartz tube reactor loaded with 100 mg of sample (PID Eng&Tech). Before loading, the catalysts were pressed in pellets, crushed and sieved to yield particles with a size between 500 and 710 μm. The NOx concentration (ppm) was measured with a Signal Model 4000VM NOx Analyzer (signal instrument). The samples were pretreated in a flow of 12.5% O₂/N₂ (800 mL min⁻¹) with a heating ramp of 30 °C min⁻¹ until 500 °C for 45 min and then cooled down to 100 °C, followed by 20 min isothermal period in N₂ atmosphere (700 mL min⁻¹). Afterward, the samples were exposed to a flow of 550 ppm NO or NO + 5% O₂ in N₂ (200 mL min⁻¹) for 40 min, and then purged for 2 h with a N₂ flow of 100 mL min⁻¹. Finally, for the TPD portion, the temperature was increased from 100 to 500 °C at a rate of 10 °C min⁻¹ in a N₂ atmosphere (100 mL min⁻¹).

**Catalyst testing.** The catalytic activity measurements of the catalysts in the NH₃-SCR reaction were carried out in a fixed bed quartz tube reactor loaded with 0.5 ml of sample (PID Eng&Tech). Before loading, the catalysts were pressed into pellets, crushed and sieved to obtain a fraction between 500 and 710 μm. Application-relevant catalyst particle size, space velocity, and gas composition were applied. The inlet NOx composition was set on pure NO to avoid higher conversions coming from the "fast SCR" mechanism when NO₂ is present. The total flow was maintained at 1000 ml/min, and the reaction condition corresponds to GHSV of 120,000 hr⁻¹. The flow rate of gases was controlled using Bronkhorst mass flow controllers. A Controlled Evaporation and Mixing system (CEM) from Bronkhorst was used for evaporation of the required H₂O in the gas feed before entering the reactor. The inlet gas stream contained 450 ppm NO, 500 ppm NH₃, 5% O₂, 5% H₂O, and N₂ balance. A MultiGas™ 2030 FTIR Continuous Gas Analyzer was used to analyze the inlet and outlet gas compositions (NO, NO₂, NH₃, N₂O). The catalytic tests were performed in the temperature range of 150–500 °C (with an interval of 50 °C) at ambient pressure. NO conversion and N₂O selectivities were calculated under steady-state conditions. The SCR activity (NO conversion) and N₂O selectivity are calculated as follows:

$$NO\ conversion\ (\%) = \frac{[NO]_{in} - [NO]_{out}}{[NO]_{out}} \times 100 \quad (1)$$

$$N_2O\ selectivity\ (\%) = \frac{2[N_2O]_{out}}{[NO_x]_{in} + [NH_3]_{in} - [NO_x]_{out} - [NH_3]_{out}} \times 100 \quad (2)$$

Where [NH₃]ₙ, [NOₓ]ₙ, [NH₃]ₒᵤₜ, [NOₓ]ₒᵤₜ, and [N₂O]ₒᵤₜ were the concentrations of NH₃ and NOₓ(including NO and NO₂) at the inlet and those at the outlet.

Surface-specific activities were calculated by normalization of the activities by the specific surface area obtained from the N₂ physisorption isotherms (BET method).

## Data availability

The data supporting the findings of this article are available in the paper and in the Supplementary Information. Additional data are available from the corresponding author on reasonable request.

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

## Acknowledgements

The authors thank the financial support from by Umicore N.V. The research was supported by the resources and facilities provided by King Abdullah University of Science and Technology. Polina Lavrik is also acknowledged for her support in the electron microscopy measurements. The authors also acknowledge the KAUST Imaging and Characterization Core Lab. We thank Sandra Ramirez-Cherbuy for the scientific illustrations of catalysts.

## Author contributions

L.E.G. and J.R.M. conceived, coordinated the research, designed the experiments, and analyzed the data. L.R.E. and A.S. synthesized the catalyst materials and characterized them by TPD and BET. L.R.E. and A.S. performed the catalytic activity measurements. S.O.-C. performed and analyzed the electron microscopy and XAS measurements with the assistance of A.A.-T. and J-L.H. A.S., and M.N.H. performed and analyzed the XPS measurements. L.R.E., A.S. performed the XRD measurements and P.P.P. conducted the analysis. C.I.Q.S. performed the Raman and NO and NO+O₂ experiments and analysis. J.R.-M. wrote the manuscript with the assistance of L.E.G., A.S., L.R.E., and F.W. All authors discussed the results and commented on the manuscript.

## Competing interests

The authors declare no competing interests.
