## [Peer Review File · Nature Communications]

Title: Unraveling the Structure and Role of Mn and Ce for Nix Reduction in Application-Relevant CatalystsREVIEWER COMMENTS

Reviewer #1 (Remarks to the Author):

In this manuscript, the author synthesized a series of Mn, Ce, and Ti mixed-oxide catalysts and tried to figure out the role of the Mn and Ce in the low-temperature SCR reaction, which is interesting and important. However, after carefully reviewing, several serious defects were found. Therefore, I do not think that this paper presents convincing results that justify its publication in Nature Communication. Specific comments are listed below.

1. Line 45, "catalyst operating at low temperatures are imperative..."
2. Line 67, "the Mn-Ce electronic interaction decrease the activity of Mn SPECIES..."
3. Although the author has proved the ternary catalysts have an amorphous phase and Mn and Ce are homogeneously distributed in the bulk and surface by XPS, ICP, TEM. It is still hard to tell the environment of the Mn and Ce in the different samples is the same, which leads to the question: what are the active sites of the MnCeTi ternary catalysts? Do they have the same structure?
4. The author claimed that Ce has a significant impact on specific surface areas of the ternary catalysts. However, if we compare samples that have the same amount of Ce but different amounts of Mn, we can also get the conclusion that Mn has a significant impact on the surface areas of the samples.
5. Fig6, the surface-specific activities were calculated at 150 oC. However, some of the conversions of NO were very high, which is far away from the kinetic region.
6. The oxidation of Mn was influenced by adding different amounts of Ce. How to exclude the effect of this on activity?

Reviewer #2 (Remarks to the Author):

General Comments: In this manuscript, it provides new insight into the understanding of the effect of Ce addition on the Mn catalysts for low temperature NH₃-SCR application, where Ce just acts as a structural promotor to increase catalyst surface area and dilute the MnO_x active sites to suppress the formation of N₂O. It is an interesting work and large number of experimental tests were provided. However, this article is not suggested to be published at this stage, since there are still lots of issues need to be addressed as follows:

- (1) The oxidation states of Mn in different determined based on XPS deconvolution method may not very accurate due to the complexity on the analysis of the Mn (2p_{3/2}) spectra and the strong interactions between MnO_x and CeO_x. Moreover, the interaction of Mn-Ce would also make the determination of oxidation states more complex. For instance, the peaks attributions due to the reduction of Mn-O-Ce and Mn-O-Mn. XANES characterizations were suggested to be performed for selected samples.
- (2) In Fig. 6, Surface-specific activities vs. the Mn content were provided. As Ce addition will decrease the surface Mn density, Mn surface density should be used as the abscissa to evaluate the effect of Ce addition on the intrinsic reaction rate.

(3) In the NH₃-SCR reaction process, the active sites are subjected to the dinuclear acid-redox site [Nature Communications (2020) 11:1532], not only the redox sites, where the adsorption of reactants on the catalyst surface also play an important role. The effects on adsorption of reactants should be also discussed with Ce addition.

(4) The method used to assess the implication of the different MnO_x species on catalytic performance as shown in Fig. 10 were not proper, since the Mn surface species (Mn⁴⁺, Mn³⁺, Mn²⁺) did not varies individually in different samples. The catalytic performances with near fixed surface Mn content could be a choice.

(5) The authors also compared the data collected from the previous studies to prove the conclusion that the no positive effect of the addition of Ce on the specific Mn activity (Table S8). However, there are lots of problems in analysis of the data in the references. For instance, in ref.2, the reaction rate vs. Mn molar ratio decreased with the reduction of Ce. In ref. 3, there are no changes in BET surface area for different samples. In ref. 4, 7, 12, the NO_x conversion for most of samples was close to 100%, the related comparisons in reaction rate is not proper.

(6) As for the analysis of N₂O formation are not main goal of this work, the relative discussion and results could be moved the supporting information part. In my opinion, the decreased redox properties of Mn-O-Ce compared to Mn-O-Mn would be possibly the main reason for enhance N₂ selectivity with suppressing the NO oxidation and/or over-oxidation of NH₃.

Reviewer #1 (Remarks to the Author):

In this manuscript, the author synthesized a series of Mn, Ce, and Ti mixed-oxide catalysts and tried to figure out the role of the Mn and Ce in the low-temperature SCR reaction, which is interesting and important. However, after carefully reviewing, several serious defects were found. Therefore, I do not think that this paper presents convincing results that justify its publication in Nature Communication. Specific comments are listed below.

1. Line 45, “catalyst operating at low temperatures are imperative...”

We thank the reviewer for the comment, we have changed the phase and now it is as follows: “catalysts operating at low temperatures are imperative...”

2. Line 67, “the Mn-Ce electronic interaction decrease the activity of Mn SPECIES...”

We have corrected and added “the Mn-Ce electronic interaction decrease the activity of Mn oxide species...”

3. Although the author has proved the ternary catalysts have an amorphous phase and Mn and Ce are homogeneously distributed in the bulk and surface by XPS, ICP, TEM. It is still hard to tell the environment of the Mn and Ce in the different samples is the same, which leads to the question: what are the active sites of the MnCeTi ternary catalysts? Do they have the same structure?

We thank the reviewer for the interesting comment. We agree that understanding the structure of the active sites is extremely difficult as the structure is completely amorphous. However, our plot of the normalized NO activity with the Mn content shows a linear relationship and therefore points that Mn oxides are the most active species (see Fig. 6 in the main manuscript). With respect to the structure, our results shows that binary MnTi catalysts are more active than the ternary MnCeTi. To have a better understanding on the structure, x-ray absorption measurements of selected samples were performed and are shown in Fig. 1. Qualitative assessment of the FT-EXAFS spectra, in Fig. 1c, show a main peak at about 1.42 Å attributed to Mn-O scattering paths and several peaks between 2.0 and 5 Å assigned mainly to Mn-Mn paths with some minor contributions from Mn-O scattering and some multiple scattering processes. The peak area corresponding to the Mn-Mn scattering paths are clearly larger for the binary $\text{Mn}_{0.35}\text{Ce}_{0.00}\text{Ti}_{0.65}$ indicating that the later materials has the best long-range order among the three composition (while still appearing amorphous by PXRD). These results are also in line with our high-resolution TEM-EDX analysis that shows that the binary Mn-Ti catalysts have a layer of MnOx on the surface of TiO_2 , therefore the MnOx species are in close proximity in the binary MnTi catalyst. In sheer contrast, the

$\text{Mn}_{0.37}\text{Ce}_{0.04}\text{Ti}_{0.60}$ and $\text{Mn}_{0.30}\text{Ce}_{0.19}\text{Ti}_{0.51}$ materials show a stronger local disorder, which is coming from the well mixing of the different metal oxides. These new results provide additional information about the structure and local environment of the Mn species in the binary MnTi and ternary MnCeTi catalysts. The x-ray absorption results are included in the revised version of the manuscript and Fig. 5 has been modified to include the spectra.

Figure 1. Mn K-edge XAS spectra for three selected catalyst compositions, $\text{Mn}_{0.35}\text{Ce}_{0.00}\text{Ti}_{0.65}$, $\text{Mn}_{0.37}\text{Ce}_{0.04}\text{Ti}_{0.60}$, $\text{Mn}_{0.30}\text{Ce}_{0.19}\text{Ti}_{0.51}$: (a) XANES, (b) EXAFS, (c) FT-EXAFS spectra.

4. The author claimed that Ce has a significant impact on specific surface areas of the ternary catalysts. However, if we compare samples that have the same amount of Ce but different amounts of Mn, we can also get the conclusion that Mn has a significant impact on the surface areas of the samples.

Our claims are based on the strong effect that additions of Ce have on catalyst's specific surface area. More specifically, small additions of Ce has a drastic impact on textural properties as can be clearly observed in Fig. 2. For example, fixing the Mn composition at around 30%, for binary MnTi the BET surface areas are around $100 \text{ m}^2/\text{g}$. In the case of small additions of Ce, in the range of 3-19 mol%, surface areas are doubled. In the case of Mn, we do not observe such a stronger effect in the range of compositions investigated. For example, fixing Ce composition around 50%, the binary CeTi has a BET surface area of $91 \text{ m}^2/\text{g}$. Additions of small amounts of Mn do have a negative effect and $\text{Mn}_{0.07}\text{Ce}_{0.56}\text{Ti}_{0.37}$, $\text{Mn}_{0.11}\text{Ce}_{0.48}\text{Ti}_{0.41}$, $\text{Mn}_{0.17}\text{Ce}_{0.40}\text{Ti}_{0.43}$ samples have surface areas of 62, 66, and $81 \text{ m}^2/\text{g}$, respectively. When Ce is around 30%, there are samples with similar ($\text{Mn}_{0.08}\text{Ce}_{0.34}\text{Ti}_{0.58}$, $\text{Mn}_{0.13}\text{Ce}_{0.31}\text{Ti}_{0.56}$, $\text{Mn}_{0.21}\text{Ce}_{0.25}\text{Ti}_{0.54}$, $\text{Mn}_{0.17}\text{Ce}_{0.40}\text{Ti}_{0.43}$, $\text{Mn}_{0.39}\text{Ce}_{0.31}\text{Ti}_{0.30}$ with 114, 86, 117, 81, 98 m^2/g , respectively) or higher surface area ($\text{Mn}_{0.15}\text{Ce}_{0.24}\text{Ti}_{0.61}$, $\text{Mn}_{0.07}\text{Ce}_{0.24}\text{Ti}_{0.69}$ with 157 and 140, respectively) when Mn is added. Summarizing,

with the number of samples investigated herein, there is no clear indication that Mn has also an structural effect on the samples.

Figure 2. Ternary diagram showing the influence of catalyst composition on BET surface area

5. Fig6, the surface-specific activities were calculated at 150 oC. However, some of the conversions of NO were very high, which is far away from the kinetic region.

We fully agree with the reviewer, some of the activities are quite high and are definitely out of the kinetic regime. First, we do not claim that this is a kinetic study, otherwise we should have made sure that the conversion level are, at least, below 5-10%. However, we still believe that this is a relevant study as in case of deviation from kinetic regime, the measured kinetic activities would be underestimated. Since the highest activities (see Fig. 6) and conversions (see Supplementary Table S8) are coming from the binary MnTi catalysts, the message of the manuscript would keep the same, reinforcing that the addition of Ce do not have a positive effect in the intrinsic activity of Mn oxide species.

6. The oxidation of Mn was influenced by adding different amounts of Ce. How to exclude the effect of this on activity?

We have already discussed the effect of the Mn oxidation states with the addition of Ce in the manuscript. We mentioned that “the increase in the Mn²⁺ species in

the samples containing Ce could, to a certain extent, explain the lower specific activity of those samples". Additionally, our TPR profiles (Fig. 5b) also show that the addition of Ce are shifting the reduction peaks of Mn^{4+} and Mn^{3+} to higher temperatures, indicating that other interactions of Mn^{4+} and Mn^{3+} with Ce species can be responsible for the lower specific activity. Overall, all these effects are caused by the addition of Ce to the catalyst formulation resulting in a decrease in the intrinsic activity of Mn species, which is our main claim in the manuscript.

Reviewer #2 (Remarks to the Author):

General Comments: In this manuscript, it provides new insight into the understanding of the effect of Ce addition on the Mn catalysts for low temperature NH_3 -SCR application, where Ce just acts as a structural promotor to increase catalyst surface area and dilute the MnO_x active sites to suppress the formation of N_2O . It is an interesting work and large number of experimental tests were provided. However, this article is not suggested to be published at this stage, since there are still lots of issues need to be addressed as follows:

(1) The oxidation states of Mn in different determined based on XPS deconvolution method may not very accurate due to the complexity on the analysis of the Mn (2p_{3/2}) spectra and the strong interactions between MnO_x and CeO_x . Moreover, the interaction of Mn-Ce would also make the determination of oxidation states more complex. For instance, the peaks attributions due to the reduction of Mn-O-Ce and Mn-O-Mn. XANES characterizations were suggested to be performed for selected samples.

We fully agree on the complexity of the XPS deconvolution to unravel the different Mn oxidation states. Because of that, we follow a rigorous method described in the main manuscript and in the supplementary information. The method is based on a) fitting with multiple Gaussian-Lorentzian components per oxidation state, b) the measurement of standards based on pure MnO , Mn_2O_3 and MnO_2 oxides and c) a comparison with previously reported results. To ensure that our deconvolution method is correct, we calculated average oxidation states by XPS and TPR and they were comparable. To reinforce our findings, the referee advices were followed and XANES characterization of selected samples were performed. The XANES spectra are plotted in fig. 1a. For the oxidation state calculations, our results were compared to other Mn-oxide standards. The comparison with manganese oxide standards which are available in crystalline form is certainly not ideal – especially when attempting linear combination fitting – but can still provide information towards a quantitative assessment of the Mn oxidation state (Fig. 3). To perform

the following work, we used Mn spectra references (Mn, MnO, Mn₃O₄, Mn₂O₃, MnOOH and MnO₂) distributed with the Hephæstus software.¹ The pre-edge energy position and the absorption threshold energy suggests that Mn is mainly in its 3+ oxidation state in the Mn_{0.35}Ce_{0.00}Ti_{0.65} composition. However, the two post-edge peaks suggest the additional presence of some amorphous MnO₂ in the solid. The average oxidation state for this sample is thus slightly higher than 3 (XPS: 3.1 and H₂-TPR: 3.25). For the Mn_{0.37}Ce_{0.04}Ti_{0.60} composition, its XANES fingerprint is quite similar to the hausmannite phase (Mn₃O₄) except that all of the spectral features appears broadened due to its amorphous nature. Considering the stoichiometry of the Mn₃O₄ phase, the average Mn oxidation state is by definition 2.66 (XPS: 2.77 and H₂-TPR: 2.6). Lastly, the Mn_{0.30}Ce_{0.19}Ti_{0.51} catalyst seems mostly composed of the same amorphous Mn₃O₄ phase with the additional presence of Mn₂O₃. In this case, the expected average Mn oxidation state should be between 2.66 and 3 (XPS: 2.75 and H₂-TPR: 3.1). In conclusion, the XANES results are in alignment with previous XPS and H₂-TPR results and are included in the revised manuscript.

Figure 3: Comparison of Mn K-edge XANES spectra for three selected catalyst compositions (Mn_{0.35}Ce_{0.00}Ti_{0.65}, Mn_{0.37}Ce_{0.04}Ti_{0.60}, Mn_{0.30}Ce_{0.19}Ti_{0.51}) with the spectra of most relevant Mn crystalline standards (MnO, Mn₃O₄, Mn₂O₃, MnO₂)

(2) In Fig. 6, Surface-specific activities vs. the Mn content were provided. As Ce addition will decrease the surface Mn density, Mn surface density should be used as the abscissa to evaluate the effect of Ce addition on the intrinsic reaction rate.

We fully understand the reviewer's point. Indeed, plotting intrinsic activities as a function of Mn surface density will be very insightful and the most adequate. In this manuscript we are reluctant to do so due to the fact that we have only measured surface Mn densities in selected samples by XPS. For the other MnCeTi ternary samples, we assume that the bulk and surface compositions are very similar, which is a good approximation considering the XPS results of selected samples and the observation of an amorphous phase in the XRD patterns. However, since we have not measured Mn surface densities for all catalysts by XPS, we prefer to be rigorous and plot values that were experimentally measured. Summarizing, for the sake of being rigorous, we prefer to keep the figure as it is.

(3) In the NH₃-SCR reaction process, the active sites are subjected to the dinuclear acid-redox site [Nature Communications (2020) 11:1532], not only the redox sites, where the adsorption of reactants on the catalyst surface also play an important role. The effects on adsorption of reactants should be also discussed with Ce addition.

The referee's comment is appreciated. To shed some light in this, we have performed additional experiments in selected samples by performing the adsorption of NO and NO+O₂ followed by a temperature-programmed desorption (Fig. 4). Quantification of NO adsorption, in table 1, shows the same trend as the activity measurements. However, in the case of the NO+O₂ adsorption (table 2), the sample with small amounts of Ce (Mn_{0.37}Ce_{0.04}Ti_{0.60}) has a larger adsorption of NO. Within the ternary catalysts, NO conversions at 150 °C are similar (41.9 and 38.8 % for the Mn_{0.37}Ce_{0.04}Ti_{0.60} and Mn_{0.30}Ce_{0.19}Ti_{0.51} samples, respectively) but the NO adsorption capacity for the ternary sample with the highest amount of Ce is always around half of the capacity for the ternary with the lowest Ce content. Overall, we can conclude that the adsorption of NO and NO+O₂ cannot fully describe the catalytic performance. These experiments are included in the revised manuscript and the following paragraph has been also included: "The effect of the adsorption of NO and NO+O₂ on catalytic performance was also investigated by temperature-programmed techniques in selected samples with similar Mn content (Supplementary Fig. 15). Quantification during NO experiments shows a higher adsorption of NO for the binary MnTi sample. In the case of the NO+O₂ adsorption measurements, the sample with a small amount of Ce has a larger amount of NO adsorbed. The lack of a clear trend evidence that the capacity of NO and NO+O₂ adsorption cannot fully describe the catalytic performance."

With respect to NH_3 , the effect of the adsorption of such molecule on N_2O selectivity was already discussed in the manuscript. The amount of ammonia adsorbed was plotted as a function of the amount of Ce (see Supplementary Fig. 14). From these results, the role of the acid sites in the N_2O formation was excluded.

Figure 4. Temperature-programmed desorption profiles of the $\text{Mn}_{0.35}\text{Ce}_{0.00}\text{Ti}_{0.65}$, $\text{Mn}_{0.37}\text{Ce}_{0.04}\text{Ti}_{0.60}$, $\text{Mn}_{0.39}\text{Ce}_{0.19}\text{Ti}_{0.51}$ samples after a) NO and b) $\text{NO}+\text{O}_2$ adsorption experiments.

Table 1. Amount of adsorbed NO during NO adsorption measurement.

Sample	NO adsorbed (μl)	NO adsorbed ($\mu\text{l m}^{-2} \text{g}^{-1}$)
$\text{Mn}_{0.35}\text{Ce}_{0.00}\text{Ti}_{0.65}$	132.28	1.13
$\text{Mn}_{0.37}\text{Ce}_{0.04}\text{Ti}_{0.60}$	111.94	0.56
$\text{Mn}_{0.39}\text{Ce}_{0.19}\text{Ti}_{0.51}$	47.43	0.20

Table 2. Amount of adsorbed NO during $\text{NO}+\text{O}_2$ adsorption measurement.

Sample	NO adsorbed (μl)	NO adsorbed ($\mu\text{l m}^{-2} \text{g}^{-1}$)
$\text{Mn}_{0.35}\text{Ce}_{0.00}\text{Ti}_{0.65}$	336.19	2.87
$\text{Mn}_{0.37}\text{Ce}_{0.04}\text{Ti}_{0.60}$	382.52	1.91
$\text{Mn}_{0.39}\text{Ce}_{0.19}\text{Ti}_{0.51}$	221.05	0.91

(4) The method used to assess the implication of the different MnOx species on catalytic performance as shown in Fig. 10 were not proper, since the Mn surface species (Mn⁴⁺, Mn³⁺, Mn²⁺) did not vary individually in different samples. The catalytic performances with near fixed surface Mn content could be a choice.

Indeed, a comprehensive assessment of the role of each MnOx species is challenging. From our measurements, we can conclude that increasing concentrations of Mn²⁺ do not have a positive effect in the specific NO activity (see Supplementary Fig. 14 b). This is also in line with previous findings². For that reason, we stated in the manuscript that “there are no direct evidences that the Mn²⁺ species are promoting catalyst activity”. Looking at samples with near constant Mn content would be an ideal approach, but that would require the synthesis of new catalytic materials and subsequent characterization. Since describing the role of the Mn oxidation state in the reaction is not the main goal of the manuscript and that would require a major experimental effort, we decided to explore this in detail in a follow-up project.

(5) The authors also compared the data collected from the previous studies to prove the conclusion that there is no positive effect of the addition of Ce on the specific Mn activity (Table S8). However, there are lots of problems in analysis of the data in the references. For instance, in ref.2, the reaction rate vs. Mn molar ratio decreased with the reduction of Ce. In ref. 3, there are no changes in BET surface area for different samples. In ref. 4, 7, 12, the NOx conversion for most of samples was close to 100%, the related comparisons in reaction rate is not proper.

We fully agree with the reviewer, some of the data from literature are not optimal and they seem to lack some details that are crucial for the analysis. However, our goal was to make an unbiased analysis of all the available data. Based on that analysis, the reported results seem to agree with our findings.

(6) As for the analysis of N₂O formation is not the main goal of this work, the relative discussion and results could be moved to the supporting information part. In my opinion, the decreased redox properties of Mn-O-Ce compared to Mn-O-Mn would be possibly the main reason for enhanced N₂ selectivity with suppressing the NO oxidation and/or over-oxidation of NH₃.

This is a valid comment and we have also mentioned the referee's hypothesis as an additional effect to decrease the N₂O formation during the reaction. Although we also agree that the N₂O formation is not the main goal of the manuscript, we would

prefer to keep this section in the manuscript to have a complete picture of the effect of Ce on activity and selectivity.

Reviewer #3 (Remarks to the Author):

In this manuscript, the authors have investigated the role of Ce on the performance of Mn-based catalysts in NH₃-SCR reaction. They reported that the intrinsic activity of the Mn active sites is not positively affected by Ce species in intimate contact at lower temperatures. Based on the surface area normalized activity results, they concluded that the advantageous effect of Ce is textural, and it increases the catalyst surface area and therefore the total number of active sites. Although the presented results are interesting, the manuscript needs a major revision before it can be accepted.

1) The authors mentioned as “The low temperature NH₃-SCR performance of all the catalysts was investigated under relevant conditions encountered in a car exhaust”. The authors just used “NO” stream to perform the NH₃-SCR reaction. However, I don’t think that a car exhaust just emits NO without producing NO₂. Then, how can the reaction conditions represent a car exhaust?

The conditions were set to mimic the N₂, O₂, H₂O, and NO_x composition of a real car exhaust. Indeed, a real combustion engine contains some NO₂. The reason avoid NO₂ in the feed stream relates to the triggering of the “fast SCR” mechanism when NO₂ is present. This has been proven to be at an order of magnitude faster than the standard SCR (with just NO) at low temperatures³⁻⁵. To clarify this, we have included the following text in the experimental section: “Application-relevant catalyst particle size, space velocity and gas composition were applied. The inlet NO_x composition was set on pure NO to avoid higher conversions coming from the “fast SCR” mechanism when NO₂ is present”.

2) How did the authors measure the specific activity of the samples? The authors should have to provide the detailed procedure for the calculation of the specific activity.

The details about the calculations of the specific activity have been included in the method section of the manuscript. In the method section, the following paragraph has been included:

“Surface-specific activities were calculated by normalization of the activities by the specific surface area obtained from the N₂ physisorption isotherms (BET method).”

3) Page 6, lines 100-102 it is stated; “In sheer contrast to the binary systems, the diffraction patterns of most of the ternary systems show featureless diffraction patterns indicating the amorphous nature of the samples (Supplementary Fig. 3)”. It would be good to provide the Raman spectra of the samples as it might be useful to know the structure of the samples.

We have followed the referee’s suggestion and Raman spectra of selected samples are included in the manuscript. Fig. 5 shows the Raman spectra of selected samples. The spectra show a band at around 600 cm⁻¹ which can be related to a Mn-O stretching mode. In general, there is a correlation with the band position and the oxidation state of Mn. Typically, MnO₂ polymorphs pyrolusite and ramsdellite have distinctive peaks located at around 665 and 650 cm⁻¹, respectively, bixbyite (Mn₂O₃) at 580 or 650-700 cm⁻¹, depending on sources, and manganosite (MnO) at 520-535 cm⁻¹.⁶⁻⁸ In our data, there is a clear trend shift of the band to lower wavenumbers with the addition of Ce (see table 1), which means that there is a reduction in the Mn oxidation state with the addition of Ce, which is in line with our previous XPS, H₂-TPR data, and recent XANES data. These results are described in the main manuscript. The experimental details, figure and deeper analysis of the data are included in the supplementary information.

Figure 5. Raman spectra of selected binary and ternary samples.

Table 3. Position of the main bands observed in the Raman spectra of selected samples

Sample	Band position (cm ⁻¹)		
S15-Mn _{0.35} Ce _{0.0} Ti _{0.65}	276.3	410.6	608.9
S24-Mn _{0.37} Ce _{0.04} Ti _{0.60}	276.3	404.6	602.9
S28-Mn _{0.39} Ce _{0.19} Ti _{0.51}	264.0	----	561.2

4) Page 6, lines 102-104; “Only a few samples with high Ti (Mn_{0.08}Ce_{0.13}Ti_{0.79}) or Ce (Mn_{0.07}Ce_{0.55}Ti_{0.37} and Mn_{0.11}Ce_{0.48}Ti_{0.41}) content display reflections from anatase TiO₂ or fluorite CeO₂, respectively”. However, the authors did not provide the XRD of Mn_{0.07}Ce_{0.55}Ti_{0.37} and Mn_{0.11}Ce_{0.48}Ti_{0.41} samples.

We thank the reviewer for spotting this. We have added the missing XRD patterns in the Supplementary Fig. 4.

5) Page 10, lines 151-152; “The catalyst materials have a clear type IV isotherm, with an H1 loop characteristic of mesoporous coming from the agglomeration of the metal-oxide nanoparticles”. The authors should provide the isotherms of the materials. This statement also needs literature support. Moreover, it is necessary to provide the pore volume and pore size of the catalysts.

The adsorption isotherms of all samples have been included in Supplementary Fig. 7. Additionally, pore volumes and pore sizes have been included in the supplementary information of the manuscript (Supplementary Table S3). We apologize for the mistake in the manuscript, the isotherms are more H2 or H3 type. We have corrected that in the main manuscript and a reference has been included. We wrote the following sentence:

“The catalyst materials have a clear type IV isotherm, with an H2 or H3 loop characteristic of mesoporous coming from the agglomeration of the metal-oxide nanoparticles”

6) Page 15, lines 239-242; “The plots in supplementary Fig. 11. shows that NO activities at 150 oC are correlated to the amount of evolved N₂O during NH₃-TPD experiments. pointing to the direct key role of the active oxygen in overall reaction mechanism at low temperature”. The authors should have to revise this sentence.

The reviewer is thanked for the comment. We have revised the sentence and wrote the following sentence:

“The plots in supplementary Fig. 11. shows that NO activities at 150 °C are correlated to the amount of N₂O evolved during the NH₃-TPD experiments, which points to the direct role of the active oxygen in the overall reaction mechanism at low temperature.”

7) Page 16-17, lines 259-261; “Therefore, we propose that Ce is not improving the catalytic properties of Mn species at low reaction temperatures and that the only promotional effect is purely structural due to an increase in catalyst surface area”.

This is incorrect as the authors found that Ce acts as a textural promoter for Mn catalysts.

We claim that Ce is not improving the intrinsic redox properties of Mn oxide species, as often claimed in the literature. We agree that the phrase in the manuscript is not completely accurate and we have changed it to the following:

“Therefore, we propose that Ce is not improving the intrinsic catalytic properties of Mn species at low reaction temperatures and that the only promotional effect of Ce is purely structural due to an increase in catalyst surface area.”

8) Page 18, lines 302-303; “amount of ammonia adsorbed, was plotted as a function of Ce content (Supplementary Fig. 13). The total acidity of all the samples was also plotted in the ternary diagram in Supplementary Fig. 14”.

Correct the Figure numbers.

We have corrected the figure numbers.

9) In Fig. 8, assign colour for the atoms in molecules participated in the mechanism.

The color code is explained now in the figure caption.

10) Particle size obtained from TEM images could support the BET results more credible.

The electron microscopy images show that the binary (supplementary Fig. 4) and ternary (Fig. 2) samples have a very heterogeneous particle size and random distribution. Considering such heterogeneous structure, the extremely local information obtained by electron microscopy and the lack of analysis for all samples, we prefer to fully rely on BET results for the surface-area analysis. Additionally, we have corrected the mistake in the main manuscript about the type of isotherm, which could be the referee's reason to back up the N₂ physisorption measurements with electron microscopy.

11) N₂ selectivity is also an important indicator, which should be given in the manuscript.

The N₂ selectivities at 150 °C have been included together with the other catalytic performance data in Supplementary Table S8.

12) The reviewer feels that the provided characterizations are not enough to deeply understand the role of Ce on the Mn catalysts. So, some in situ and advanced experiments should be supplemented.

Based on all reviewers comments, additional data to understand the role of Ce on the Mn samples have been included. Based on the XAS and Raman data, the effect of Ce on decreasing the oxidation state of Mn and on the well mixing of the metal oxides were confirmed. Additional results based on the adsorption of NO and NO + O₂ were also included, suggesting that the adsorption of such gases is not determining catalytic performance. All those new experiments have been included in the manuscript.

13) It would be good if the authors can discuss the role of Ti on the Mn catalysts.

As we discussed in the manuscript, the addition of a third metal oxide component, which could be TiO₂, is critical for the formation of a ternary amorphous phase: "In sheer contrast to the binary systems, the diffraction patterns of most of the ternary systems show featureless diffraction patterns indicating the amorphous nature of the samples (Supplementary Fig. 3)... this suggest that the third metal component facilitates the formation of a ternary amorphous phase, which is the first step on obtaining a homogeneous mixed oxide". In the specific case of Ti oxide, it also has

the role of support in the binary MnTi samples, as can be observed in the electron microscopy images (see supplementary Fig. 5). This was also discussed in the manuscript: “We observed nanosized and crystalline TiO₂ particles decorated with layers of amorphous material. High-angle annular dark-field (HAADF) STEM imaging and elemental mappings computed from energy-dispersive X-ray spectroscopy (EDX) data, presented in Supplementary Fig. 5, reveal that Mn is well dispersed on the TiO₂ particles.” For the binary MnTi and ternary MnTiCe, there is a clear trend in the specific activities with the Mn content, therefore any catalytic effect of Ti oxide can be ruled out.

14) Some recent papers on Mn-based NH₃-SCR catalysis are missing and should be cited by the authors in the revised version of this paper.

Applied Surface Science 534 (2020) 147592; Molecular Catalysis 501 (2021) 111376; Applied Catalysis B: Environmental 287 (2021) 119939; Applied Surface Science 508 (2020) 144694.

The suggested references have been included in the main manuscript.

References

1. Ravel, B.; Newville, M., ATHENA, ARTEMIS, HEPHAESTUS: data analysis for X-ray absorption spectroscopy using IFEFFIT. *J Synchrotron Radiat* **2005**, *12* (Pt 4), 537-41.
2. Liu, C.; Shi, J.-W.; Gao, C.; Niu, C., Manganese oxide-based catalysts for low-temperature selective catalytic reduction of NO_x with NH₃: A review. *Applied Catalysis A: General* **2016**, *522*, 54-69.
3. Nova, I.; Ciardelli, C.; Tronconi, E.; Chatterjee, D.; Bandl-Konrad, B., NH₃-NO/NO₂ chemistry over V-based catalysts and its role in the mechanism of the Fast SCR reaction. *Catalysis Today* **2006**, *114* (1), 3-12.
4. Kato, A.; Matsuda, S.; Kamo, T.; Nakajima, F.; Kuroda, H.; Narita, T., Reaction between nitrogen oxide (NO_x) and ammonia on iron oxide-titanium oxide catalyst. *The Journal of Physical Chemistry* **1981**, *85* (26), 4099-4102.
5. Zhang, S.; Zhang, B.; Liu, B.; Sun, S., A review of Mn-containing oxide catalysts for low temperature selective catalytic reduction of NO_x with NH₃: reaction mechanism and catalyst deactivation. *RSC advances* **2017**, *7* (42), 26226-26242.
6. Post, J. E.; McKeown, D. A.; Heaney, P. J., Raman spectroscopy study of manganese oxides: Tunnel structures. *American Mineralogist* **2020**, *105* (8), 1175-1190.
7. Bernardini, S.; Bellatreccia, F.; Della Ventura, G.; Sodo, A., A Reliable Method for Determining the Oxidation State of Manganese at the Microscale in Mn Oxides via Raman Spectroscopy. *Geostandards and Geoanalytical Research* **2021**, *45* (1), 223-244.
8. Bernardini, S.; Bellatreccia, F.; Casanova Municchia, A.; Della Ventura, G.; Sodo, A., Raman spectra of natural manganese oxides. *Journal of Raman Spectroscopy* **2019**, *50* (6), 873-888.

REVIEWERS' COMMENTS

Reviewer #1 (Remarks to the Author):

The authors answered the reviewers' questions well. This article is ready for publication.

Reviewer #3 (Remarks to the Author):

The authors have addressed all my concerns very well.

Please accept as is.

REVIEWERS' COMMENTS

Reviewer #1 (Remarks to the Author):

The authors answered the reviewers' questions well. This article is ready for publication.

Reviewer #3 (Remarks to the Author):

The authors have addressed all my concerns very well.
Please accept as is.

We are delighted to read that the referees are satisfied with the revised version submitted and that they consider that the manuscript is ready for publication. We deeply appreciated all referee comments as they have been critical to bring the revised version to a substantial higher level.